

# Association between serum ferritin and uric acid levels and nonalcoholic fatty liver disease in the Chinese population

Fangli Zhou[1],[*], Xiaoli He[2],[*], Dan Liu[1], Yan Ye[3], Haoming Tian[1] and Li Tian[4]

[1] West China Hospital, Sichuan University, Department of Endocrinology, Chengdu, Sichuan, China
[2] West China Hospital, Sichuan University, Department of Outpatient, Chengdu, Sichuan, China
[3] West China Fourth Hospital, Sichuan University, Department of Gastroenterology, Chengdu, Sichuan, China
[4] West China Hospital, Sichuan University, Laboratory of Endocrinology and Metabolism, Department of Endocrinology, Chengdu, Sichuan, China
* These authors contributed equally to this work.

## ABSTRACT

**Background:** The prevalence of nonalcoholic fatty liver disease (NAFLD) is increasing worldwide. Accumulating evidence suggests that serum ferritin and uric acid (UA) are strongly associated with the risk of NAFLD, but no consensus has been reached.

**Objective:** We sought to demonstrate the association between serum ferritin, UA levels, and NAFLD risk in a large cohort study.

**Methods:** We separated 2,049 patients into non-NAFLD and NAFLD groups. The NAFLD group had four subgroups based on serum ferritin and four subgroups based on UA quartile levels. We used binary logistic regression to evaluate the correlation between serum ferritin, UA, and NAFLD. Additionally, an area under the curve (AUC) of receiver operating characteristic analysis (ROC) was used to predict the diagnostic value of combined serum ferritin and UA for NAFLD.

**Results:** Serum ferritin and UA levels were higher in the NAFLD group compared with the non-NAFLD group. Serum lipid and liver transaminase concentrations were elevated with the increase of serum ferritin and UA. The logistic regression results showed an independent correlation between serum ferritin, UA, and NAFLD. In the NAFLD group, the AUC value of serum ferritin and UA was 0.771.

**Conclusions:** Increased serum ferritin and UA levels are independent risk factors for NAFLD. Increased serum UA is a stronger risk factor for NAFLD than elevated serum ferritin. Serum ferritin and UA can be important predictors of NAFLD risk.

## INTRODUCTION

Non-alcoholic fatty liver disease (NAFLD) is the most prevalent liver disease in the world. It is characterized by excess intracellular fat in the absence of excessive alcohol consumption (*Wang et al., 2022*; *Jung et al., 2019*). NAFLD is a disease that progresses in

Corresponding authors
Haoming Tian, hmtian999@126.com
Li Tian, litian1981228@gmail.com

stages starting with simple fatty liver (steatosis), nonalcoholic steatohepatitis (NASH), fibrosis, and cirrhosis, which can lead to liver failure or even hepatocellular carcinoma (*Yang et al., 2022*; *Rostoker et al., 2019*). The incidence of NAFLD is increasing and epidemiological studies have shown that it affects more than 25% of the adult population worldwide (*Wong et al., 2018*). In China, the prevalence of NAFLD has increased to 29.2% (*Wang et al., 2022*). A survey of 7,152 employees in Shanghai showed that approximately 38.17% had NAFLD (*Hu et al., 2012*). Additionally, the number of children and adolescents with NAFLD is rapidly increasing (*Nobili et al., 2019*).

NAFLD is strongly associated with obesity, type 2 diabetes (T2DM), insulin resistance (IR), and metabolic syndrome (*Branisso et al., 2022*). It is a major cause of liver disease worldwide (*Younossi et al., 2018*) and is associated with an increased risk of cardiovascular events (*Deprince, Haas & Staels, 2020*). Ferritin is a protein that stores iron and its role in iron homeostasis has been studied (*Kernan & Carcillo, 2017*). Ferritin is an acute phase reactant, so different amounts of it in serum can reflect the severity of inflammation (*Kernan & Carcillo, 2017*; *Kowdley et al., 2012*). Nearly 33% of hyperferritinemia cases could indicate an overload of iron in the liver (*Branisso et al., 2022*). Higher serum ferritin is an independent risk factor for NASH and advanced fibrosis in patients with NAFLD (*Kowdley et al., 2012*). According to the results of a 16-year follow-up survey, serum ferritin is also associated with increased mortality in patients with NAFLD (*Hagström et al., 2016*). Serum ferritin is a non-invasive and easily available biomarker of NAFLD (*Yang et al., 2022*; *Giacomazzi et al., 2022*). However, there is still uncertainty about its association with NAFLD (*Modares Mousavi et al., 2018*; *Du et al., 2017*) therefore, its role should be investigated in a large-scale study (*Yang et al., 2022*; *Giacomazzi et al., 2022*). Many previous studies have suggested that hyperferritinemia is not only associated with NAFLD and systemic inflammation, but also correlates with the histologic severity of NAFLD and with the presence of NASH (*Wang et al., 2022*).

UA is the ultimate product of purine metabolism in the body. Many studies have reported that UA is associated with an increased incidence of NAFLD (*Yuan et al., 2015*; *Wei et al., 2020*; *Wang et al., 2022*). NAFLD is related to excessive fructose intake. UA stimulates fructokinase, which promotes fructose metabolism and fat deposition in liver cells (*Lanaspa et al., 2012*). This may account for the increased UA levels in NAFLD (*Heda et al., 2021*). In contrast, some studies have shown a significant negative correlation between UA and NAFLD (*Zhou et al., 2016*). Considering the inconsistent results of the association between serum UA and NAFLD, more studies are needed to characterize their association (*Tang et al., 2022*). Although liver biopsies are a gold standard for NAFLD diagnosis, they are not routinely performed because of their invasiveness and high cost. Studies have shown that increased serum ferritin and UA are involved in the pathogenesis of NAFLD. However, studies that combine these biomarkers to evaluate the risk of NAFLD are rare. Therefore, this study sought to explore the correlation between serum ferritin, UA, and NAFLD risk.

## MATERIALS AND METHODS

### Study participants

We enrolled 1,103 patients with NAFLD and 1,080 patients without NAFLD in August 2015 at the Health Examination Center of West China Hospital using a cross-sectional study design. The inclusion criteria were as follows: (1) fatty liver found by ultrasonography and laboratory biochemical parameter examination; (2) being a resident of Sichuan Province; (3) aged 15 to 84 years old. The exclusion criteria were as follows: (1) men with an alcohol intake of more than 140 g/week; (2) women with alcohol consumption of more than 70 g/week; (3) presence of other liver illnesses such as hepatitis B or hepatitis C infection; (4) malignancies; (5) pregnancy; (6) long-term use of estrogens; (7) tamoxifen, or corticosteroids; (8) body mass index (BMI) less than 16; (9) incomplete data. The Ethics Committee of West China Hospital, Sichuan University, approved the trial protocol (IRB approval number 2021-1455). All participants gave informed consent in the form of written consent. Our study was registered in the Chinese Clinical Trial Registry (ChiCTR2100049091).

### Ultrasonography and definition of NAFLD

Hepatic ultrasonography was done by trained technicians who were blinded to the study design and made the diagnosis of NAFLD by ultrasound scan. We used a high-resolution B-mode ultrasonic probe (IU22; Philips, Amsterdam, Netherlands) equipped with a 7.5 MHz linear array to measure the fatty liver. During the scan, participants were in the supine position with their right arms raised above their heads. The images were photographed and recorded by two sonographers who were unaware of the goals of the study and blinded to the laboratory results. Fatty liver was defined as a diffuse increase of fine echoes in the liver parenchyma compared with the kidney or spleen parenchyma (increased liver echogenicity at ultrasound examination reflects the degree of steatosis but not of fibrosis in asymptomatic patients with mild or moderate abnormalities of liver transaminases). Radiologists determined the presence of fatty liver using live images.

### Data collection and clinical measurement

Questionnaires and electronic medical records were used to collect demographic information, such as age, gender, anthropometric data, smoking and drinking habits, medical history, and family history of diabetes mellitus and hypertension. Body mass index (BMI) was calculated by dividing weight in kilograms by height in meters squared (kg/m$^2$). Waist circumference (WC) was measured at the minimum circumference between the iliac crest and the rib cage; the hip-circumference was measured at the maximum circumference over the buttocks using a non-stretching tape measure. Both circumference measurements were recorded to the nearest 0.1 cm. The waist-to-hip ratio (WHR) was calculated by dividing the waist circumference by the hip circumference. The Homeostasis Model Assessment of Insulin Resistance (HOMA-IR) was used to evaluate insulin resistance. The standard formula for this measurement is HOMA-IR = (glucose × insulin)/22.5.
Venous blood samples were collected from participants after they fasted for 8 h or more. These samples were used to measure biochemical indicators, including triglycerides (TG), total cholesterol (TC), low-density lipoprotein cholesterol (LDL-C), high-density lipoprotein cholesterol (HDL-C), hemoglobin (Hb), aspartate aminotransferase (AST), alanine aminotransferase (ALT), alkaline phosphatase (ALP), γ-glutamyl transpeptidase (GGT), fasting plasma glucose (FPG), UA, alpha fetoprotein (AFP), and creatinine. All values were assessed using automated, standardized equipment from the Clinical Laboratory of West China Hospital. Radioimmunoassay (Beijing North Institute of Biological Technology) was used to test serum ferritin and fasting insulin.

## Participants subgroup classification

The participants were divided into the control group and the NAFLD group; the NAFLD patients were then further categorized in accordance with their ferritin quartiles (Q1, ≤76.9 ng/mL; Q2, 77.0–150.8 ng/mL; Q3, 150.9–275.6 ng/mL; Q4, ≥275.7 ng/mL); UA quartiles (Q1, ≤305 μm/L; Q2, 306–368 μm/L; Q3, 369–435 μm/L; Q4, ≥436 μm/L).

## Statistics analysis

Data were analyzed with SPSS Statistics for Windows, version 25.0 (IBM Corp., Armonk, NY, USA). Continuous variables were presented as mean ± standard deviation (normal distribution) or median and interquartile range (non-normal distribution).
An independent sample $t$-test, Mann–Whitney $U$ test or the Kruskal–Wallis $H$ test was used for the analysis of baseline characteristics and laboratory test parameters between the NAFLD and non-NAFLD groups, where appropriate. Pearson correlation analysis was performed to analyze the correlation between serum ferritin levels, UA level, and other clinical variables. A one-way ANOVA was used to assess the difference in the continuous variables among the quartile groups. Binary logistic regression was used to explore the independent risk factors for NAFLD. The combined diagnostic accuracy of serum ferritin and UA was determined using the ROC curve with calculated the AUC. Results were considered statistically significant when $P$ was less than 0.05.

# RESULTS

## Baseline patient characteristics

After the inclusion and exclusion criteria were applied, 2,049 participants remained. Of these, 1,017 were in the non-NAFLD group (49.6%) and 1,032 in the NAFLD group (50.4%). The average age, BMI, WC, hip circumference, and WHR of participants in the NAFLD group were significantly higher than those in the non-NAFLD group.

Compared with the non-NAFLD group, ALT, AST, ALP, GGT, TG, TC, LDL-C, Hb, AFP, UA, creatinine, ferritin, FPG, insulin, and HOMA-IR were significantly increased (all $P < 0.01$ except $P < 0.05$ for AFP) while the HDL-C value significantly decreased in the NAFLD group ($P < 0.01$) (Table 1).

**Table 1 Baseline clinical characteristics of the study participants.**

| | Total (N = 2,049) | Non-NAFLD (N = 1,017) | NAFLD (N = 1,032) | P value |
|---|---|---|---|---|
| Age (yrs) | 41.5 ± 12.1 | 39.6 ± 12.1 | 43.4 ± 11.7 | <0.001 |
| BMI (kg/m$^2$) | 24.5 ± 3.5 | 22.5 ± 2.9 | 26.5 ± 2.8 | <0.001 |
| WC (cm) | 82.8 ± 10.5 | 76.3 ± 8.9 | 89.1 ± 7.7 | <0.001 |
| Hip circumference (cm) | 95.3 ± 6.1 | 92.5 ± 5.3 | 98.1 ± 5.4 | <0.001 |
| WHR | 0.87 ± 0.08 | 0.82 ± 0.07 | 0.91 ± 0.06 | <0.001 |
| SBP (mmHg) | 119.5 ± 25.4 | 114.0 ± 15.0 | 124.9 ± 31.7 | <0.001 |
| DBP (mmHg) | 75.3 ± 10.7 | 71.7 ± 9.8 | 79.0 ± 10.3 | <0.001 |
| ALT (IU/L) | 27.0 (18, 42) | 20 (14, 29) | 37 (26, 55) | <0.001 |
| AST (IU/L) | 25.0 (20, 31) | 22 (19, 27) | 28 (23, 36) | <0.001 |
| ALP (IU/L) | 75.6 ± 20.6 | 71.5 ± 19.2 | 79.6 ± 21.2 | <0.001 |
| GGT (IU/L) | 25 (15, 45) | 17 (12, 26) | 38 (24, 64) | <0.001 |
| TG (mmol/L) | 1.5 (1.0, 2.3) | 1.1 (0.8, 1.6) | 2.1 (1.5, 2.8) | <0.001 |
| TC (mmol/L) | 4.9 ± 0.9 | 4.7 ± 0.9 | 5.1 ± 0.9 | <0.001 |
| HDL-C (mmol/L) | 1.4 ± 0.4 | 1.6 ± 0.4 | 1.2 ± 0.3 | <0.001 |
| LDL-C (mmol/L) | 2.6 ± 0.7 | 2.5 ± 0.7 | 2.8 ± 0.7 | <0.001 |
| Hb (g/L) | 151.1 ± 15.7 | 146.9 ± 16.2 | 155.2 ± 13.9 | <0.001 |
| AFP (ng/mL) | 3.7 ± 2.2 | 3.5 ± 2.4 | 3.8 ± 2.0 | <0.05 |
| UA (μmol/L) | 374.1 ± 98.1 | 332.1 ± 84.1 | 415.5 ± 93.2 | <0.001 |
| Creatinine (μmol/L) | 77.5 ± 16.3 | 74.2 ± 16.5 | 80.7 ± 15.4 | <0.001 |
| Ferritin (ng/mL) | 150.4 (76.9, 275.6) | 115 (53, 194) | 209.5 (131.8, 319.4) | <0.001 |
| FPG (mmol/L) | 5.4 ± 1.1 | 5.1 ± 0.7 | 5.6 ± 1.4 | <0.001 |
| FINS (μU/mL) | 14.3 ± 6.8 | 12.6 ± 5.1 | 16.0 ± 7.8 | <0.001 |
| HOMA-IR | 3.4 ± 2.0 | 2.9 ± 1.3 | 4.0 ± 2.3 | <0.001 |

**Note:**

BMI, body mass index; WC, waist circumference; WHR, waist circumference to hip circumference ratio; SBP, systolic blood pressure; DBP, diastolic blood pressure; ALT, alanine aminotransferase; AST, aspartate aminotransferase; ALP, alkaline phosphatase; GGT, glutamine transaminase; TG, triglyceride; TC, total cholesterol; HDL-C, high-density lipoprotein cholesterol; LDL-C, low-density lipoprotein cholesterol; Hb, hemoglobin; AFP, alpha fetoprotein; UA, uric acid; FPG, fasting plasma glucose; FINS, fasting serum insulin; HOMA-IR, homeostasis model assessment of insulin resistance; NAFLD, nonalcoholic fatty liver disease

### Baseline characteristics of participants with NAFLD based on the serum ferritin quartile

NAFLD subjects were divided into four subgroups based on their serum ferritin quartile levels (Q1, ≤76.9 ng/mL; Q2, 77.0–150.8 ng/mL; Q3, 150.9–275.6 ng/mL; and Q4, ≥275.7 ng/mL). Table 2 shows that BMI, WC, diastolic blood pressure (DBP), ALT, AST, GGT, TG, Hb, creatinine, FPG, and insulin levels all increased with higher serum ferritin concentrations. However, age, hip circumference, systolic blood pressure (SBP), ALP, insulin, and HOMA-IR had no significant difference between the ferritin subgroups.

### Baseline characteristics of participants with NAFLD based on the serum UA quartile

Quartiles of serum UA were also in four subgroups (Q1, ≤305 μm/L; Q2, 305–368 μm/L; Q3, 369–435 μm/L; and Q4, ≥436 μm/L). The incidence of NAFLD showed an increasing trend with higher levels of serum UA, and the incidences in the first to fourth subgroups

**Table 2 Comparison of different indices in participants with NAFLD based on ferritin quartiles.**

| | Q1 (≤76.9 ng/mL) (N = 130) | Q2 (77.0–150.8 ng/mL) (N = 229) | Q3 (150.9–275.6 ng/mL) (N = 299) | Q4 (≥275.7 ng/mL) (N = 374) | P value |
|---|---|---|---|---|---|
| Age (yrs) | 42.7 ± 11.1 | 44.8 ± 13.5 | 43.9 ± 11.9 | 42.4 ± 10.5 | NS |
| BMI (kg/m²) | 25.6 ± 3.0 | 26.4 ± 2.7 | 26.6 ± 2.7 | 26.8 ± 2.8 | <0.001 |
| WC (cm) | 85.2 ± 8.9 | 88.5 ± 7.2 | 89.5 ± 7.0 | 90.5 ± 7.7 | <0.001 |
| Hip circumference (cm) | 97.1 ± 5.8 | 97.8 ± 5.7 | 98.4 ± 5.3 | 98.3 ± 5.3 | NS |
| WHR | 0.88 ± 0.07 | 0.91 ± 0.06 | 0.91 ± 0.05 | 0.92 ± 0.05 | <0.001 |
| SBP (mmHg) | 123.2 ± 18.0 | 124.6 ± 15.7 | 124.4 ± 16.2 | 126.0 ± 47.9 | NS |
| DBP (mmHg) | 77.1 ± 10.4 | 78.5 ± 10.7 | 78.9 ± 10.4 | 80.0 ± 9.7 | <0.05 |
| ALT (IU/L) | 28 (19, 46) | 31 (22, 46) | 38 (27, 53) | 43 (29, 64) | <0.001 |
| AST (IU/L) | 25 (20, 33) | 26 (22, 32) | 29 (23, 34) | 31 (24, 40) | <0.001 |
| ALP (IU/L) | 78.6 ± 26.7 | 79.8 ± 19.5 | 79.6 ± 21.0 | 80.0 ± 20.3 | NS |
| GGT (IU/L) | 28 (17, 46) | 33 (22, 58) | 38(25, 57) | 45 (28, 77) | <0.001 |
| TG (mmol/L) | 1.7 (1.2, 2.3) | 2.0 (1.4, 2.9) | 2.1 (1.4, 2.8) | 2.2 (1.6, 3.1) | <0.001 |
| TC (mmol/L) | 4.8 ± 0.8 | 5.1 ± 1.0 | 5.0 ± 0.8 | 5.1 ± 0.9 | <0.01 |
| HDL-C (mmol/L) | 1.3 ± 0.3 | 1.3 ± 0.3 | 1.2 ± 0.3 | 1.2 ± 0.3 | <0.001 |
| LDL-C (mmol/L) | 2.6 ± 0.7 | 2.8 ± 0.8 | 2.8 ± 0.7 | 2.8 ± 0.7 | <0.05 |
| Hb (g/L) | 142.0 ± 15.3 | 154.1 ± 13.7 | 157.6 ± 11.8 | 158.7 ± 12.2 | <0.001 |
| AFP (ng/mL) | 3.4 ± 1.5 | 3.7 ± 1.9 | 3.7 ± 1.6 | 4.1 ± 2.4 | <0.01 |
| UA (μmol/L) | 5.5 ± 1.3 | 5.5 ± 1.5 | 5.6 ± 1.3 | 5.7 ± 1.4 | NS |
| Creatinine (μmol/L) | 349.0 ± 74.8 | 399.3 ± 83.3 | 421.8 ± 87.3 | 443.5 ± 96.0 | <0.001 |
| Ferritin (ng/mL) | 68.2 ± 14.3 | 80.0 ± 17.7 | 82.4 ± 13.7 | 84.2 ± 13.2 | <0.001 |
| FPG (mmol/L) | 49.4 (29.2, 69.0) | 129.4 (108.2, 141.5) | 206.8 (166.4, 243.9) | 349.2 (309.6, 398.1) | <0.001 |
| FINS (μU/mL) | 15.3 ± 6.3 | 15.6 ± 7.3 | 15.7 ± 7.5 | 16.6 ± 8.7 | NS |
| HOMA-IR | 3.8 ± 2.1 | 3.8 ± 2.1 | 3.9 ± 2.3 | 4.3 ± 2.6 | NS |

**Note:**

BMI, body mass index; WC, waist circumference; WHR, waist circumference to hip circumference ratio; SBP, systolic blood pressure; DBP, diastolic blood pressure; ALT, alanine aminotransferase; AST, aspartate aminotransferase; ALP, alkaline phosphatase; GGT, glutamine transaminase; TG, triglyceride; TC, total cholesterol; HDL-C, high-density lipoprotein cholesterol; LDL-C, low-density lipoprotein cholesterol; Hb, hemoglobin; AFP, alpha fetoprotein; UA, uric acid; FPG, fasting plasma glucose; FINS, fasting serum insulin; HOMA-IR, homeostasis model assessment of insulin resistance; NAFLD, nonalcoholic fatty liver disease; NS, no significance.

were 5.27%, 10.88%, 15.03%, and 19.18%, respectively (Table 3). Meanwhile, the results showed that the subjects with a higher concentration of UA were more likely to be young and obese. The DBP, ALT, AST, GGT, TG, Hb, creatinine, and ferritin concentration were higher in the highest (Q4) quartile group. TC, LDL-C, AFP, FPG, insulin along with HOMA-IR showed no significant difference between the subgroups.

## Relationship between serum ferritin, UA levels, and other factors in participants with NAFLD

Serum ferritin and UA levels were significantly positively related to BMI, WC, hip circumference, WHR, DBP, ALT, AST, GGT, TG, TC, Hb, AFP, UA, creatinine, insulin, and HOMA-IR, while significantly negatively related to age and HDL-C. Additionally, the FPG was significantly negatively associated with serum UA levels but significantly positively associated with serum ferritin levels (Table 4).

**Table 3  Comparison of different indices in participants with NAFLD based on UA quartiles.**

| | Q1 (≤305 µm/L) (N = 108) | Q2 (306–368 µm/L) (N = 223) | Q3 (369–435 µm/L) (N = 308) | Q4 (≥436 µm/L) (N = 393) | P value |
|---|---|---|---|---|---|
| Age (yrs) | 47.2 ± 12.4 | 47.1 ± 11.5 | 42.0 ± 11.2 | 41.3 ± 11.4 | <0.001 |
| BMI (kg/m²) | 25.0 ± 2.4 | 25.7 ± 2.4 | 26.6 ± 2.8 | 27.3 ± 2.9 | <0.001 |
| WC (cm) | 84.2 ± 7.8 | 86.5 ± 7.5 | 89.6 ± 7.4 | 91.5 ± 7.1 | <0.001 |
| Hip circumference (cm) | 95.7 ± 5.0 | 96.3 ± 4.7 | 98.4 ± 5.4 | 99.4 ± 5.5 | <0.001 |
| WHR | 0.88 ± 0.06 | 0.90 ± 0.06 | 0.91 ± 0.05 | 0.92 ± 0.05 | <0.001 |
| SBP (mmHg) | 119.9 ± 16.8 | 124.6 ± 17.1 | 123.5 ± 13.9 | 127.4 ± 47.3 | NS |
| DBP (mmHg) | 73.2 ± 9.9 | 77.8 ± 10.5 | 79.3 ± 9.6 | 81.0 ± 10.1 | <0.001 |
| ALT(IU/L) | 28 (19, 43) | 32 (23, 46) | 37 (26, 55) | 42 (29, 62) | <0.001 |
| AST (IU/L) | 26 (20, 33) | 27 (22, 33) | 28 (23, 35) | 30 (25, 39) | <0.001 |
| ALP (IU/L) | 78.5 ± 26.5 | 79.2 ± 20.1 | 80.3 ± 20.2 | 79.7 ± 20.9 | NS |
| GGT (IU/L) | 25 (15, 46) | 30 (21, 48) | 37 (24, 58) | 47 (30, 76) | <0.001 |
| TG (mmol/L) | 1.6 (1.2, 2.2) | 1.8 (1.4, 2.5) | 2.0 (1.4, 2.8) | 2.4 (1.7, 3.2) | <0.001 |
| TC (mmol/L) | 4.9 ± 0.9 | 5.0 ± 0.9 | 5.0 ± 0.8 | 5.2 ± 0.9 | NS |
| HDL-C (mmol/L) | 1.4 ± 0.3 | 1.3 ± 0.3 | 1.2 ± 0.3 | 1.2 ± 0.3 | <0.001 |
| LDL-C (mmol/L) | 2.7 ± 0.7 | 2.8 ± 0.7 | 2.8 ± 0.7 | 2.8 ± 0.7 | NS |
| Hb (g/L) | 142.4 ± 15.9 | 152.5 ± 14.1 | 156.9 ± 13.0 | 159.0 ± 11.3 | <0.001 |
| AFP (ng/mL) | 3.5 ± 1.7 | 3.8 ± 1.9 | 3.7 ± 1.8 | 3.9 ± 2.2 | NS |
| UA (µmol/L) | 265.2 ± 28.6 | 340.9 ± 18.6 | 402.9 ± 19.9 | 509.0 ± 61.8 | <0.001 |
| Creatinine (µmol/L) | 67.6 ± 15.4 | 73.5 ± 13.7 | 81.2 ± 12.2 | 88.0 ± 14.5 | <0.001 |
| Ferritin (ng/mL) | 110.3 (62.9, 235.2) | 159.5 (103.1, 297.3) | 210.9 (132.4, 316.7) | 258.6 (155.9, 347.4) | <0.001 |
| FPG (mmol/L) | 5.9 ± 2.0 | 5.8 ± 1.5 | 5.5 ± 1.1 | 5.5 ± 1.3 | NS |
| FINS (µU/mL) | 15.2 ± 6.3 | 15.1 ± 6.1 | 16.0 ± 8.7 | 16.7 ± 8.3 | NS |
| HOMA-IR | 4.0 ± 1.9 | 4.0 ± 2.1 | 4.0 ± 2.5 | 4.2 ± 2.5 | NS |

Note:
BMI, body mass index; WC, waist circumference; WHR, waist circumference to hip circumference ratio; SBP, systolic blood pressure; DBP, diastolic blood pressure; ALT, alanine aminotransferase; AST, aspartate aminotransferase; ALP, alkaline phosphatase; GGT, glutamine transaminase; TG, triglyceride; TC, total cholesterol; HDL-C, high-density lipoprotein cholesterol; LDL-C, low-density lipoprotein cholesterol; Hb, hemoglobin; AFP, alpha fetoprotein; UA, uric acid; FPG, fasting plasma glucose; FINS, fasting serum insulin; HOMA-IR, homeostasis model assessment of insulin resistance; NAFLD, nonalcoholic fatty liver disease; NS, no significance.

## Binary logistic regression analysis of the association between ferritin and UA with NAFLD

Binary logistic regression analysis (backward:LR) was conducted to adjust for confounding factors that may contribute to NAFLD to ultimately confirm whether ferritin and UA were independent risk factors.

As shown in Table 5, the analysis found that gender, BMI, hip circumference, ALT, AST, FPG, creatinine, TG, TC, and LDL-C were risk factors for NAFLD. The results also showed that the serum ferritin and UA levels were independently associated with the occurrence of NAFLD.

The ORs for NAFLD from ferritin-Q2 to ferritin-Q4 were 1.135 (95% CI [0.757–1.702]), 1.366 (0.886–2.105), and 1.756 (1.110–2.778), respectively. Similarly, compared with the Q1, the serum UA level was significantly associated with NAFLD risk in the Q2 to Q4 subgroups ($P < 0.05$; $P < 0.01$; $P < 0.001$). The risk of NAFLD in the Q3-UA and Q4-UA

**Table 4 Correlation analysis of parameters based on serum ferritin and UA levels among participants with NAFLD.**

| Ferritin | R | P | UA | R | P |
|---|---|---|---|---|---|
| Age | −0.041 | 0.187 | | −0.202 | <0.001 |
| BMI | 0.134 | <0.001 | | 0.287 | <0.001 |
| WC | 0.212 | <0.001 | | 0.326 | <0.001 |
| Hip circumference | 0.085 | 0.01 | | 0.265 | <0.001 |
| WHR | 0.217 | <0.0011 | | 0.221 | <0.001 |
| SBP | 0.028 | 0.375 | | 0.067 | <0.05 |
| DBP | 0.097 | 0.01 | | 0.212 | <0.001 |
| ALT | 0.267 | <0.001 | | 0.246 | <0.001 |
| AST | 0.223 | <0.001 | | 0.190 | <0.001 |
| ALP | 0.017 | 0.587 | | 0.011 | 0.712 |
| GGT | 0.206 | <0.001 | | 0.210 | <0.001 |
| TG | 0.147 | <0.001 | | 0.190 | <0.001 |
| TC | 0.099 | 0.01 | | 0.090 | <0.01 |
| HDL-C | −0.147 | <0.001 | | −0.226 | <0.001 |
| LDL-C | 0.056 | 0.072 | | 0.047 | 0.132 |
| Hb | 0.263 | <0.001 | | 0.309 | <0.001 |
| AFP | 0.124 | <0.001 | | 0.059 | 0.059 |
| UA | 0.305 | <0.001 | | | |
| Creatinine | 0.253 | <0.001 | | 0.483 | <0.001 |
| Ferritin | | | | 0.305 | 0.001 |
| FPG | 0.066 | <0.05 | | −0.098 | <0.01 |
| FINS | 0.063 | <0.05 | | 0.064 | <0.05 |
| HOMA-IR | 0.081 | <0.01 | | 0.016 | 0.603 |

**Note:**

BMI, body mass index; WC, Waist circumference; WHR, waist circumference to hip circumference ratio; SBP, systolic blood pressure; DBP, diastolic blood pressure; ALT, alanine aminotransferase; AST, aspartate aminotransferase; ALP, alkaline phosphatase; GGT, glutamine transaminase; TG, triglyceride; TC, total cholesterol; HDL-C, high-density lipoprotein cholesterol; LDL-C, low-density lipoprotein cholesterol; Hb, hemoglobin; AFP, alpha fetoprotein; UA, uric acid; FPG, fasting plasma glucose; FINS, fasting serum insulin; HOMA-IR, homeostasis model assessment of insulin resistance; NAFLD, nonalcoholic fatty liver disease.

subgroups increased almost two (OR, 2.088; 95% CI [1.351-3.226]) and three times (OR, 3.173; 95% CI [1.943–5.182]), respectively.

## Diagnostic value of combined serum ferritin and UA for NAFLD

The above results suggested an association between serum ferritin, UA, and the risk of NAFLD. In this study, the ROC analysis was used to analyze the predictive accuracy of combined serum ferritin and UA for NAFLD diagnosis. The AUC value of combined ferritin and UA was 0.771 (95% CI [0.751–0.791]) with a sensitivity of 71% and a specificity of 70%, which indicated its good diagnostic value for NAFLD (Fig. 1).
**Table 5 Binary logistic regression analysis of the association between serum ferritin and UA in participants with NAFLD.**

| Variables | OR | 95% CI | P | OR | 95% CI | P |
|---|---|---|---|---|---|---|
| Ferritin (Quartile) | | | | UA (Quartile) | | |
| Q1 | Ref | | | Ref | | |
| Q2 | 1.135 | [0.757–1.702] | 0.540 | 1.487 | [1.003–2.206] | <0.05 |
| Q3 | 1.366 | [0.886–2.105] | 0.158 | 2.088 | [1.351–3.226] | <0.01 |
| Q4 | 1.756 | [1.110–2.778] | <0.05 | 3.173 | [1.943–5.182] | <0.001 |
| Gender (male vs female) | 4.295 | [2.704–6.824] | <0.001 | 4.428 | [2.790–7.027] | <0.001 |
| BMI | 1.217 | [1.113–1.330] | <0.001 | 1.224 | [1.120–1.338] | <0.001 |
| WC | 0.612 | [0.392–0.955] | <0.05 | 0.655 | [0.424–1.013] | 0.057 |
| Hip circumference | 1.629 | [1.103–2.406] | <0.05 | 1.537 | [1.050–2.250] | <0.05 |
| ALT | 1.031 | [1.020–1.041] | <0.001 | 1.030 | [1.019–1.041] | <0.001 |
| AST | 0.984 | [0.968–0.999] | <0.05 | 0.984 | [0.969–1.000] | <0.05 |
| FPG | 1.295 | [1.112–1.508] | <0.01 | 1.304 | [1.121–1.518] | <0.01 |
| Creatinine | 0.988 | [0.978–0.999] | <0.05 | 0.990 | [0.980–1.001] | 0.072 |
| UA | 1.005 | [1.003–1.007] | <0.001 | | | |
| TG | 2.098 | [1.762–2.498] | <0.001 | 2.104 | [1.768–2.503] | <0.001 |
| TC | 0.306 | [0.192–0.488] | <0.001 | 0.306 | [0.193–0.486] | <0.001 |
| LDL-C | 4.886 | [2.788–8.563] | <0.001 | 4.982 | [2.852–8.703] | <0.001 |
| Ferritin | | | | 1.002 | [1.001–1.003] | <0.01 |

Note:
BMI, body mass index; WC, Waist circumference; ALT, alanine aminotransferase; AST, aspartate aminotransferase; FPG, fasting plasma glucose; UA, uric acid; TG, triglyceride; TC, total cholesterol; LDL-C, low-density lipoprotein cholesterol.

## DISCUSSION

A total of 2,049 participants were enrolled and the association of serum ferritin and UA with NAFLD was investigated. We found that the participants in the NAFLD group had significantly higher levels of serum ferritin and UA than those in the non-NAFLD group.

Further subgroup analysis showed that with increased serum ferritin levels, participants in the NAFLD group displayed more severe metabolic dysregulation (BMI, WC, TG, creatinine, glucose, insulin, HOMR-IR increased) and liver function abnormality (ALT, AST, and GGT increased). Furthermore, the prevalence of NAFLD was much higher in the highest ferritin quartile subgroup (18.3%) than in the lowest one (6.3%). Logistic regression analysis suggested that an elevated serum ferritin concentration was an independent risk factor of NAFLD after adjusting for various potential confounding factors. These findings are consistent with previous studies in different country populations (*Jung et al., 2019*; *Yang et al., 2022*; *Wang et al., 2022*; *Kowdley et al., 2012*).

Circulating ferritin levels reflect iron stores in the body. They are also part of the pro-inflammatory cytokine-induced inflammatory signaling cascade associated with chronic liver injury and cirrhosis (*Ruddell et al., 2009*). Both *in vivo* and *in vitro* studies have shown that hepatocytes and Kupffer cells can secrete ferritin (*Kernan & Carcillo,*

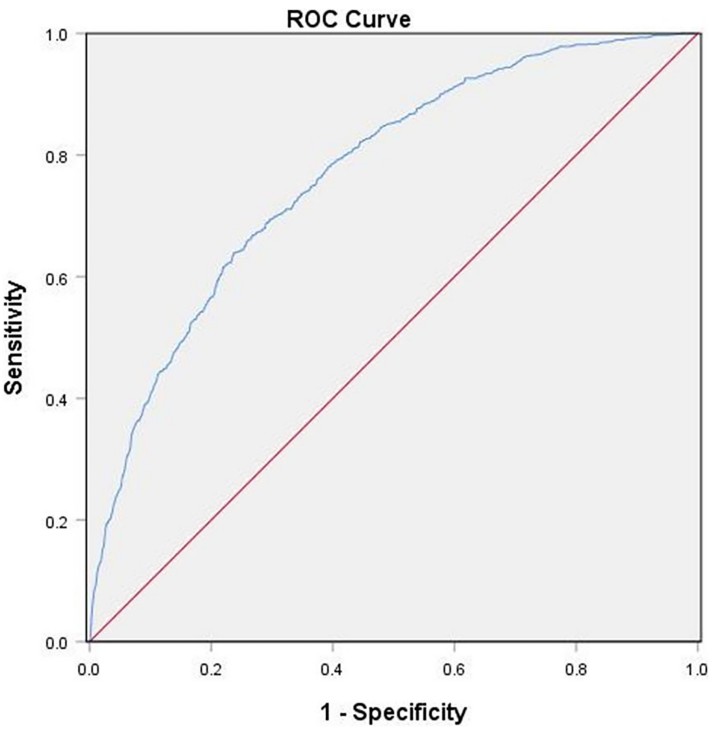

**Figure 1 ROC Serum ferritin and UA Curves for Participants With NAFLD.** The diagnostic accuracy of combined serum ferritin and UA in all participants using the ROC method. Combined serum ferritin and UA had an AUC of 0.771 (95% CI [0.751–0.791]).

2017), and inflammatory cytokines also increase ferritin expression (*Kernan & Carcillo, 2017*; *Kowdley et al., 2012*; *Pham et al., 2004*). Iron is a strong pro-oxidant that catalyzes reactive oxidative species (ROS) production and leads to cellular damage (*Bessone, Razori & Roma, 2019*). Also, higher ferritin has been observed to increase lipid peroxidation (*Aranda et al., 2016*).

We also observed that BMI and waist circumference increased proportionally with serum ferritin concentration. Waist circumference is regarded as an indicator of central obesity. BMI increase and visceral obesity are traditional risk factors for NAFLD (*Yan et al., 2021*). A higher BMI is correlated to hyperferritinemia regardless of the iron storage condition in participants (*Alam, Memon & Fatima, 2015*), along with a deteriorating NAFLD fibrosis score (*Tutunchi et al., 2021*). Obesity is closely associated with metabolic disorders and chronic inflammation, which further promote NAFLD development (*Yan et al., 2021*).

Additionally, we found that an increase in TG, TC, and LDL-C levels, especially TG, and a decrease in HDL-C levels were associated with higher serum ferritin levels, which was further confirmed by Pearson correlation analysis. Iron overload usually manifests as high ferritin and transferrin saturation that indirectly causes insulin resistance through liver injury and lipid metabolism abnormalities (*Gao et al., 2022*). *Al Akl et al. (2021)* reported a significant relationship between serum ferritin and abnormalities in lipid parameters. Also, a positive correlation between serum ferritin and TG was found in diabetic mellitus

(*Musina et al., 2020*). Moreover, a higher serum ferritin concentration was associated with the prevalence of hyperlipidemia independent of glucose metabolism disorders and metabolic syndrome in a large cross-sectional study (*Li et al., 2017*). An increase in serum TG may increase free fatty acids (FFA) and contribute to insulin resistance and B cell dysfunction. Therefore, serum ferritin may interfere with glucose metabolism *via* lipid metabolism. A study by *Hevi & Chuck (2003)* demonstrated that serum ferritin blocks the apoB secretion, which results in an elevation of TG.

The association between serum UA and the risk of NAFLD has been reported in other studies (*Wei et al., 2020*; *Wang et al., 2022*; *Zhou et al., 2016*). We found that serum UA levels were significantly elevated in NAFLD participants ($415.5 \pm 93.2$ µmol/L *vs.* $332.1 \pm 84.1$ µmol/L; $P < 0.001$). Participants with NAFLD in Q4 were younger (mean 41.3 years old) and had a higher BMI (mean 27.3), WC, and TG levels compared with Q1. The OR of the highest UA quartile ($\geq 436$ µmol/L) compared with the lowest one ($\leq 305$ µmol/L) was 3.173 (1.943–5.182). The biomarkers of liver injury, including serum ALT, AST, and GGT, especially ALT, were significantly higher in subjects with increased serum UA.

The prevalence of NAFLD is increasing rapidly alongside diabetes, obesity, and metabolic syndrome. Traditionally, these conditions are related to excess caloric intake, inadequate physical activity, and unhealthy dietary habits, especially the use of high-fructose corn syrup (HFCS), which is the primary sweetener used in soft drinks and fruit juice (*Ouyang et al., 2008*). Industrialization has changed our lifestyles greatly. Data from the National Health and Nutrition Examination Survey (NHANES) indicated that compared with older adults, young adults had the highest intake and percent of daily calories from sugar-sweetened beverages (*Rosinger et al., 2017*). The evidence demonstrates that HFCS beverage consumption is linked to weight gain and hypertriglyceridemia, which is consistent with the metabolic features of our participants based on serum UA levels (*Ouyang et al., 2008*). Fructose is catalyzed in the liver by fructokinase (ketohexokinase, KHK) that consumes adenosine-triphosphate (ATP) to phosphorylate fructose to fructose-1-phosphate. This leads to ATP depletion in the liver, which can generate UA (*Lanaspa et al., 2012*; *Ouyang et al., 2008*). Because of this, hyperuricemic people are more likely to develop fructose-induced fatty liver.

Different pathologic mechanisms are responsible for the relationship between UA and NAFLD. A study by *Lanaspa et al. (2012)* demonstrated that UA, either alone or as a by-product of fructose phosphorylation, directly stimulated hepatic lipogenesis because of enhanced mitochondrial translocation of the nicotinamide adenine dinucleotide phosphate (NADPH) oxidase isoform, NOX4. Reduced aconitase activity led to mitochondrial citrate accumulation, which eventually caused hepatic *de novo* lipogenesis and TG accumulation. The UA-induced hepatocyte fat accumulation is also involved in activating the NOD-like receptor family pyrin domain containing 3 (NLRP3) inflammasome (*Wan et al., 2016*), endoplasmic reticulum (ER) stress, and mitochondrial oxidative stress (*Hu et al., 2018*). Additionally, UA could induce hepatocyte lipid accumulation *via* regulation of the miR-149-5p/FGF21 axis (*Chen et al., 2020*).

These results suggest that serum ferritin and UA are independent risk factors for NAFLD and may be related to multiple signaling pathways. High serum UA leads to a

greater risk of NAFLD compared with high serum ferritin. Combining serum ferritin and UA has a higher predictive accuracy for NAFLD (AUC of 0.771; 95% CI [0.751–0.791]), thus indicating its good diagnostic value. Therefore, based on previous studies, we should pay attention to the presence of elevated ferritin and UA levels in patients with NAFLD. If the optimal cut-off serum ferritin level is determined, it can be used to predict NAFLD development.

There are certain limitations in our research. First, NAFLD was confirmed by ultrasonography, which is not the gold standard for diagnosis of NAFLD. A liver biopsy, which is the standard of NAFLD diagnosis, has not been widely used in clinical practice because of the inconvenience and high risk involved. Ultrasonography does not provide histologic details on the severity of liver damage. Second, this study did not include lifestyle factors such as smoking, alcohol consumption, use of sugar-sweetened beverages by participants and dietary habits that could affect regression analysis.

## CONCLUSION

Patients with NAFLD had significantly increased serum ferritin and UA levels. Increases in these levels were associated with more severe lipid metabolism disorders and liver enzyme abnormalities. Serum ferritin and UA levels were independent risk factors for NAFLD, and combining serum ferritin and UA could be a powerful biological indicator of NAFLD diagnosis.

## ACKNOWLEDGEMENTS

We thank TopEdit for its linguistic assistance during the preparation of this manuscript.

### Funding

This study was supported by the West China Hospital, Sichuan University (Grant No. 2020HXBH028 to Fangli Zhou), the China Postdoctoral Science Foundation (Grant No. 2020M670060ZX to Fangli Zhou), the National Natural Science Foundation of China (Grant No. 31300648 to Li Tian) and the Sichuan Science and Technology Program (Grant No. 2023YFSY0039 to Li Tian). The Chengdu Science and Technology program (Grant No. 2022-YF05-01497-SN to Fangli Zhou) and the National Natural Science Foundation of China Grant (No. 82200641 to Fangli Zhou) supported the APC for this article. The funders had no role in study design, data collection and analysis, decision to publish, or preparation of the manuscript.

### Grant Disclosures

The following grant information was disclosed by the authors:
Sichuan University: 2020HXBH028.
China Postdoctoral Science Foundation: 2020M670060ZX.
National Natural Science Foundation of China: 31300648.
Sichuan Science and Technology Program: 2023YFSY0039.

Chengdu Science and Technology program: 2022-YF05-01497-SN.
National Natural Science Foundation of China: 82200641.

## Competing Interests

The authors declare that they have no competing interests.

## Author Contributions

- Fangli Zhou conceived and designed the experiments, performed the experiments, analyzed the data, prepared figures and/or tables, authored or reviewed drafts of the article, and approved the final draft.
- Xiaoli He conceived and designed the experiments, performed the experiments, authored or reviewed drafts of the article, and approved the final draft.
- Dan Liu performed the experiments, analyzed the data, prepared figures and/or tables, and approved the final draft.
- Yan Ye performed the experiments, prepared figures and/or tables, and approved the final draft.
- Haoming Tian conceived and designed the experiments, authored or reviewed drafts of the article, and approved the final draft.
- Li Tian analyzed the data, prepared figures and/or tables, authored or reviewed drafts of the article, and approved the final draft.

## Human Ethics

The following information was supplied relating to ethical approvals (*i.e.*, approving body and any reference numbers):

The Ethics Committee of West China Hospital, Sichuan University, approved the trial protocol.

## Data Availability

The raw data is available in the Supplemental Files.

## Supplemental Information

Supplemental information for this article can be found online at http://dx.doi.org/10.7717/peerj.16267#supplemental-information.

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
