# Peer review of "Association between serum ferritin and uric acid levels and nonalcoholic fatty liver disease in the Chinese population"

_PeerJ, doi:10.7717/peerj.16267_

## Round 0.1 · original submission · Major Revisions

The authors should consider all the concerns raised by the reviewers. English should be rechecked for grammar/typos.

Reviewer 1 ·

Basic reporting

This is a well-written manuscript, evaluating serum uric acid and ferritin levels as biomarkers for NAFLD with a sufficient sample size.
Manuscript structure, figures, and tables conform to standard structures and are relevant.

Experimental design

The main defect of this study is incomplete data in the method section.
The authors didn't mention the design of the study. However, it seems this is an observational retrospective case-control study.
Data on the inclusion of controls are minimal. It is stated that data were derived from medical records and questionnaires in a hospital setting.
• What was the indication of liver ultrasonography in control subjects? Without data on control subjects, the design of the study is at risk of selection bias.
It should be better to select controls from the normal population, rather than those who are referred to the hospital.

• In exclusion criteria, participants who take lipid-lowering, hypouricemic, iron supplements, hepatotoxic medications, or medications that affect the development of hepatic steatosis (such as steroids, immune suppressants, anticonvulsants, or tetracycline) were not excluded. These conditions can significantly change serum uric acid or ferritin levels.

Validity of the findings

Conclusions are well stated and linked to the results. However, the rationale for performing this research is not well stated. At line 65, the authors declare that " Currently, serum ferritin is a non-invasive and easily available biomarker of NAFLD, but it should be investigated on a large-scale study"; Many previous studies suggest that not only hyperferritinemia is associated with NAFLD and systemic inflammation, but also it has been reported to correlate with histologic severity of NAFLD and with the presence of NASH (https://doi.org/10.3389/fmed.2022.934989). Nevertheless, the combination of uric acid and ferritin as biomarkers for NAFLD diagnosis is relatively a novel idea and sheds light on future studies.

Additional comments

The paper needs to be rechecked grammatically and some sentences are better to be improved with their structure. Some suggestions are mentioned below:
• At line 108, mention the waist and hip circumference scale.
• At the end of line 125, it is better to write ‘’ between serum ferritin level, UA level, and other clinical variables’’.
• At line 145, ‘’hip’’ is not complete.
• At line 158, ‘’circumstance’’ is falsely was used, should be replaced with ‘’circumference’’.
• Lines 198-199, ‘’cytokine-induced cascade inflammation signaling’’ is better to be rephrased to be better understood.
Abbreviations should be checked to be defined at first use in both abstract and main text.
At lines 64 and 65, if the association of NAFLD and serum ferritin is still conflicting, how it is a biomarker?
At line 175, ‘’difference between the second and third quartiles’’ is not true, as they were compared to the first quartile separately.
The abbreviation of HOMA-IR under the tables is not defined exactly.
• Please mention the sensitivity and specificity of AUC-ROC analysis.

In the method section:
Subgroups of NAFLD based on serum ferritin and UA quartiles are better to be defined in methods.
• In methods it was mentioned that in January 2016 the patients were enrolled, however, in raw data the enrollment dates are August 2015.
• In the first inclusion criteria, specify the ‘’other examinations’’, everything in methods should be obviously defined.
• About the second inclusion criteria, "voluntary participation" is a rule in human studies and is better not be mentioned as a criteria

Reviewer 2 ·

Basic reporting

1. Please take care when using the word “correlation”. In fact, the logistic regression was used to explore the “association” instead of “correlation”. Please only use “correlation” when correlation test was done. Please correct this in the paper.

2. In section Binary Logistic Regression Analysis of the Association between Ferritin and UA With NAFLD, line 175, “However, no significant difference between the second and third ferritin quartiles was found.”, this result cannot be obtained based on the logistic regression since the p-values reflects the Q2 vs Q1, Q3 vs Q1, Q4 vs Q1.

Experimental design

Main concerns:

1. For Statistics Analysis section, “Differences between normally distributed variables were analyzed using an independent sample t-test. The variables with non-normal distribution were compared using the Mann Whitney U test or the Kruskal Wallis H test”, please clarify what statistical method was used for multiple group comparison for the variables normally distributed. F-test are expected to be used for this case (comparison among Q1, Q2, Q3, Q4). My concern is that the authors used t-test for all above cases that I just mentioned.

2. In section Binary Logistic Regression Analysis of the Association between Ferritin and UA With NAFLD, please justify why you build the three model: Crude model, Model I, Model II. Particularly for Model I, can you please justify why you adjust these covariates? Actually, a model selection process is recommended here, e.g. backward selection. The AUC of ROC should be done for the model finally selected.

Validity of the findings

The authors explored association between serum ferritin, uric acid levels and nonalcoholic fatty liver disease in the Chinese population, which is an impressive topic. But the statistical methods need to be clarified and justified to make the findings more promising.

Additional comments

NA

---

## Round 0.2 · accepted · Accept

The authors have satisfactorily addressed the issues raised by the reviewers and greatly improved English text.

Reviewer 1 ·

Basic reporting

NA

Experimental design

NA

Validity of the findings

NA

Additional comments

The authors have changed accordingly and I have no further comments.

Reviewer 2 ·

Basic reporting

The refined manuscript provided sufficient background and literature references and had professional article structure. The figures tables and row data were shared properly. The manuscript used clear and professional English throughout.

Experimental design

For the refined manuscript, the research question were well defined and meaningful. A relatively rigorous investigation were performed based on a high technical and ethical standard. The research methods were well described with enough details and info after the paper refinement.

Validity of the findings

Since the manuscript was refined according to reviewers' comments, the methods in the refined paper made sense. So the results and conclusion were relatively promising. The underlying data was provided which is robust and statistically controlled. The conclusions were well stated and closely related to original research question.

Additional comments

I recommend to accept.